# Training Deep Neural Networks with 8-bit Floating Point Numbers

**Naigang Wang, Jungwook Choi, Daniel Brand, Chia-Yu Chen and Kailash Gopalakrishnan**
IBM T. J. Watson Research Center
Yorktown Heights, NY 10598, USA
`{nwang, choij, danbrand, cchen, kailash}@us.ibm.com`

## Abstract

The state-of-the-art hardware platforms for training Deep Neural Networks (DNNs) are moving from traditional single precision (32-bit) computations towards 16 bits of precision – in large part due to the high energy efficiency and smaller bit storage associated with using reduced-precision representations. However, unlike inference, training with numbers represented with less than 16 bits has been challenging due to the need to maintain fidelity of the gradient computations during back-propagation. Here we demonstrate, for the first time, the successful training of DNNs using 8-bit floating point numbers while fully maintaining the accuracy on a spectrum of Deep Learning models and datasets. In addition to reducing the data and computation precision to 8 bits, we also successfully reduce the arithmetic precision for additions (used in partial product accumulation and weight updates) from 32 bits to 16 bits through the introduction of a number of key ideas including chunk-based accumulation and floating point stochastic rounding. The use of these novel techniques lays the foundation for a new generation of hardware training platforms with the potential for $2 - 4\times$ improved throughput over today's systems.

## 1 Introduction

Over the past decade, Deep Learning has emerged as the dominant Machine Learning algorithm showing remarkable success in a wide spectrum of applications, including image processing [9], machine translation [20], speech recognition [21] and many others.

In each of these domains, Deep Neural Networks (DNNs) achieve superior accuracy through the use of very large and deep models – necessitating up to 100s of ExaOps of computation during training and Gigabytes of storage. Approximate computing techniques have been widely studied to minimize the computational complexity of these algorithms as well as to improve the throughput and energy efficiency of hardware platforms executing Deep Learning kernels [2]. These techniques trade off the inherent resilience of Machine Learning algorithms for improved computational efficiency. Towards this end, exploiting reduced numerical precision for data representation and computation has been extremely promising – since hardware energy efficiency improves quadratically with bit-precision.

While reduced-precision methods have been studied extensively, recent work has mostly focused on exploiting them for *DNN inference*. It has shown that the bit-width for inference computations can be successfully scaled down to just a few bits (i.e., 2-4 bits) while (mostly) preserving accuracy [3]. However, *reduced precision DNN training* has been significantly more challenging due to the need to maintain fidelity of the gradients during the back-propagation step. Recent studies have empirically shown that at least 16 bits of precision is necessary to train DNNs without impacting model accuracy [6, 16, 4]. As a result, state-of-the-art training platforms have started to offer 16-bit floating point training hardware [8, 5] with $\geq 4\times$ performance over equivalent 32-bit systems.

The goal of this paper is to push the envelope further and enable DNN training using 8-bit floating point numbers. To exploit the full benefits of 8-bit platforms, 8-bit floating point numbers are used for numerical representation of data as well as computations encountered in the forward and backward passes of DNN training. There are three primary challenges to using super scaled precision while fully preserving model accuracy (as exemplified in Fig. 1 for ResNet18 training on ImageNet dataset). Firstly, when all the operands (i.e., weights, activations, errors and gradients) for general matrix multiplication (GEMM) and convolution computations are reduced to 8 bits, most DNNs suffer noticeable accuracy degradation (e.g., Fig. 1(a)). Secondly, reducing the bit-precision of accumulations in GEMM from 32 bits (e.g., [16, 4]) to 16 bits significantly impacts the convergence of DNN training (Fig. 1(b)). This reduction in accumulation bit-precision is critically important for reducing the area and power of 8-bit hardware. Finally, reducing the bit-precision of weight updates to 16-bit floating point impacts accuracy (Fig. 1(c)) - while 32-bit weight updates require an extra copy of the high precision weights and gradients to be kept in memory, which is expensive.

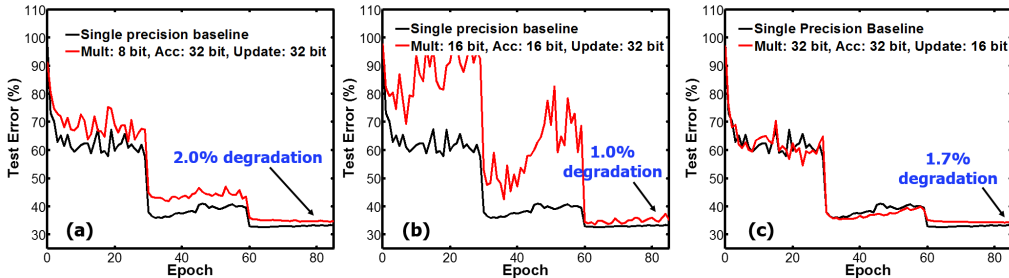

Figure 1: The challenges of **selectively** reducing training precision with (a) 8-bit representations, (b) 16-bit accumulations, and (c) 16-bit weight updates vs. $FP32$ baseline for ResNet18 (ImageNet).

In this paper, we introduce new techniques to fully overcome all of above challenges:

- Devised a new $FP8$ floating point format that, in combination with DNN training insights, allows GEMM computations for Deep Learning to work without loss in model accuracy.
- Developed a new technique called **chunk-based computations** that when applied hierarchically allows all matrix and convolution operations to be computed using only **8-bit multiplications** and **16-bit additions** (instead of 16 and 32 bits respectively).
- Applied **floating point stochastic rounding** in the weight update process allowing these updates to happen with **16 bits of precision** (instead of 32 bits).
- Demonstrated the wide applicability of the combined effects of these techniques across a suite of Deep Learning models and datasets – while fully preserving model accuracy.

The use of these novel techniques open up new opportunities for hardware platforms with $2-4\times$ improved energy efficiency and throughput over state-of-the-art training systems.

## 2   8-bit floating point training

### 2.1   Related Work

There has been a tremendous body of research conducted towards DNN precision scaling over the past few years. However, a significant fraction of this quantization research has focused around reduction of bit-width for the forward path for inference applications. Recently, precision for weights and activations were scaled down to 1-2 bits ([11, 3]) with a small loss of accuracy, while keeping the gradients and errors in the backward path as well as the weight updates in full-precision. In comparison to inference, much of the recent work on low precision training often uses much higher precision – specifically on the errors and gradients in the backward path. DoReFa-Net [22] reduces the gradient precision down to 6 bits while using 1-bit weights and 2-bit activations for training. WAGE [19] quantizes weights, activations, errors and gradients to 2, 8, 8 and 8 bits respectively. However, all of these techniques incur significant accuracy degradation ($> 5\%$) relative to full-precision models.

To maintain model accuracy for reduced-precision training, much of recent work keeps the data and computation precision in at least 16 bits. MPT [16] uses a IEEE half-precision floating point

format (16 bits) accumulating results into 32-bit arrays and additionally proposes a loss-scaling method to preserve gradients with very small magnitudes. Flexpoint [13] and DFP [4] demonstrated a format with a 16-bit mantissa and a shared exponent to train large neural networks with full-precision accuracy. The shared exponents can be adjusted dynamically to minimize overflow. However, even with **16-bit data representations**, these techniques require the partial products to be **accumulated in 32-bits** and subsequently rounded down to 16 bits for the following computation. In addition, in all cases, a 32-bit copy of the weights is maintained to preserve the fidelity of the weight update process.

In contrast, using the new ideas presented in this paper, we show that it is possible to train these networks using just 8-bit floating point representations for all of the arrays used in matrix and tensor computations – weights, activations, errors and gradients. In addition, we show that the partial products of these two 8-bit operands can be accumulated into 16-bit sums which can then be rounded down to 8 bits again. Furthermore, the master copy of the weights preserved after the weight update process can be scaled down from 32 to 16 bits. These advances dramatically improve the computational efficiency, energy efficiency and memory bandwidth needs of future deep learning hardware platforms without impacting model convergence and accuracy.

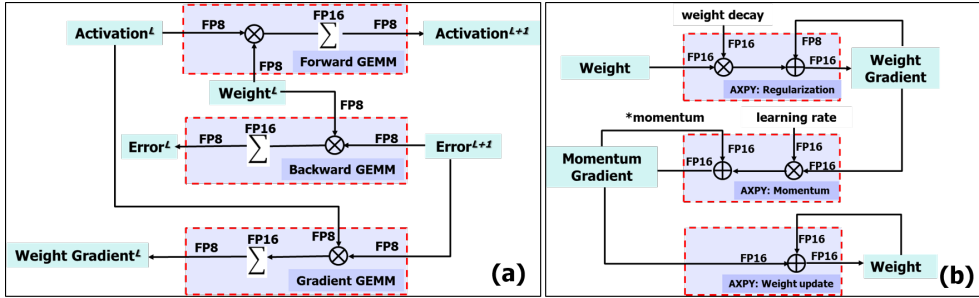

Figure 2: A diagram showing the precision settings for (a) three GEMM functions during forward and backward passes, and (b) three AXPY operations during a standard SGD weight update process.

## 2.2 New Reduced Precision Floating Point Formats: $FP8$ and $FP16$

Fig. 2 shows the precision settings for the three GEMM functions during forward and backward passes, i.e., Forward, Backward and Gradient GEMM, as well as the three vector addition (AXPY) operations during a standard stochastic gradient descent (SGD) weight update process, i.e., L2-regularization (L2-Reg), momentum gradient accumulation (Momentum-Acc), and weight update (Weight-Upd). Note that the convolution computation is implemented by first lowering the input data, followed by GEMM operations. In other words, GEMM refers to computations corresponding to both convolution (Conv) and fully-connected (FC) layers. Our 8-bit floating point number ($FP8$) has a (sign, exponent, mantissa) format of **(1, 5, 2)** bits - where the format is chosen carefully to represent weights, activations, errors and gradients used in the three GEMMs. The 16-bit floating point number ($FP16$) has a **(1, 6, 9)** format and is used for both GEMM accumulations as well as AXPY additions – where the higher (6-bit) exponent provides a larger dynamic range needed during weight updates. Both $FP8$ and $FP16$ formats are selected after in-depth studies of the data distribution in networks, focusing on balancing the representation accuracy and dynamic range. Due to limited available space, we only show results from the best formats that work reliably across a variety of deep networks/datasets. We refer to IEEE single precision as $FP32$, i.e **(1, 8, 23)**. In addition, we explore two floating point rounding modes post $FP16$ additions – nearest and stochastic rounding.

## 2.3 Floating Point Accumulation in Reduced Precision

A GEMM function involves a dot-product that may accumulate a large number of element-wise products in floating point. Since floating point addition involves right-shift of the smaller of the two operands (by the difference in exponents), it is possible that this smaller number may be truncated entirely after addition due to limited mantissa bits.

This issue of truncation in large-to-small number addition (also called "**swamping**" [10]) is known in the area of high performance computing [17], which focuses on numerical accuracy of high precision 32/64-bit floating point computations. However, in the context of deep neural networks, we find that the swamping is particularly serious when the accumulation bit-precision is reduced aggressively. When we use our $FP16$ format for accumulations, this truncation happens when the magnitude differs larger than the swamping threshold $2^{mantissa+1}$. Furthermore, swamping is exacerbated under the following conditions: 1) the accumulation is done over the values with non-zero mean (and thus the magnitude of the sum can gradually increase beyond the swamping threshold) and/or 2) some of the elements in the vector have a large magnitude (due to long tails in the distribution). These two cases cause significant accumulation errors – and is the reason why **current hardware platforms are unable to reduce accumulation precision below 32 bits**.

In this work, we demonstrate that swamping severely limits reduction in training precision and propose two novel schemes that completely overcome this limit and enable low-precision $FP8$ DNN training: chunk-based accumulation and floating point stochastic rounding.

**Chunk-based Accumulation**   The novel insight behind our proposed idea of chunk-based accumulations is to divide a long dot-product into smaller chunks (defined by the chunk length $CL$). The individual element-wise products are then added hierarchically – intra-chunk accumulations are first performed to produce partial sums followed by inter-chunk accumulations of these partial sums to produce a final dot-product value. Since the length of the additions for both intra-chunk and inter-chunk computations is reduced by $CL$, the probability of adding a large number to a small number decreases dramatically. Furthermore, chunk-based accumulation requires little additional computational overhead (unlike sorting-based summation techniques) and incurs relatively insignificant memory overheads (unlike pairwise-summation) while reducing theoretical error bounds from $O(N)$ to $O(N/CL + CL)$ where $N$ is the length of the dot product – similar to the analysis in [1].

Motivated by this chunk-based accumulation concept, we propose a reduced-precision dot-product algorithm for Deep Learning as described in Fig. 3(a). The input to the dot-product are two vectors in $FP_{mult}$ precision, which are multiplied in $FP_{mult}$ but have products accumulated in a higher precision $FP_{acc}$ in order to capture information of the intermediate sum better, e.g., $FP_{mult} = FP8$ and $FP_{acc} = FP16$. Since $FP16$ is still significantly lower than the typical bit-precision used in GPUs today for GEMM accumulation (i.e., $FP32$), we employ chunk-based accumulation to overcome swamping errors. Intra-chunk accumulation is carried out in the innermost loop of the algorithm shown in Fig. 3(a), then the sum of the chunks is further accumulated into the final sum. It should be noted that only a single additional variable is required to maintain the intra-chunk sum – thereby minimizing cost and overheads. The net impact of this remarkably simple idea is to minimize swamping and to open up opportunities for using $FP8$ for representations (and multiplications) and $FP16$ for accumulations, while matching $FP32$ baselines for additions as shown in Fig. 3(b).

**Stochastic Rounding**   Stochastic rounding is another extremely effective way of addressing the issue of swamping. Note that information loss occurs when the bit-width is reduced by rounding. As discussed before, floating point addition rounds off the intermediate sum of two aligned mantissas. Nearest rounding is a common rounding mode, but it discards information conveyed in the least significant bits (LSBs) that are rounded off. This information loss can be significant when the accumulation bit-precision is reduced into half, i.e., $FP16$, which has only 9 bits of mantissa.

Stochastic rounding is a method to capture this information loss from the discarded bits. Assume a floating point value with the larger mantissa bits for the intermediate sum, $x = s \cdot 2^e \cdot (1 + m)$ where $s$, $e$, and $m$ are sign, exponent, and mantissa for $x$, respectively. Also assume that $m$ for this intermediate sum is represented in fixed-precision with $k'$ bits, which needs to be rounded off into smaller bits, $k \leq k'$. Then, the stochastic rounding works as follows:

$$Round(x) = \begin{cases} s \cdot 2^e \cdot (1 + \lfloor m \rfloor + \epsilon) & \text{with probability } \frac{m - \lfloor m \rfloor}{\epsilon}, \\ s \cdot 2^e \cdot (1 + \lfloor m \rfloor) & \text{with probability } 1 - \frac{m - \lfloor m \rfloor}{\epsilon}, \end{cases} \tag{1}$$

where $\lfloor m \rfloor$ is the truncation of $k' - k$ LSBs of $m$, and $\epsilon = 2^{-k}$.

Note that this floating point stochastic rounding technique is mathematically different from the fixed point stochastic rounding approach that is widely used in literature [6, 11]; since the magnitude of the rounding error of the floating point stochastic rounding is proportional to the exponent value $2^e$.

Table 1: Training configuration and test error (model size) across a spectrum of networks and datasets.

| Model | CIFAR10-CNN | CIFAR10-ResNet | BN50-DNN | AlexNet | ResNet18 | ResNet50 |
|---|---|---|---|---|---|---|
| Dataset | CIFAR10 | CIFAR10 | BN50 | ImageNet | ImageNet | ImageNet |
| Minibatch Size | 128 | 128 | 256 | 256 | 256 | 256 |
| Epoch | 140 | 160 | 20 | 45 | 85 | 80 |
| $FP32$ Baseline | 17.80% (0.45MB) | 7.23% (2.81MB) | 59.33% (64.5MB) | 41.96% (432MB) | 32.57% (66.9MB) | 27.86% (147MB) |
| Our $FP8$ Training | 18.15% (0.23MB) | 7.79% (1.41MB) | 60.08% (34.5MB) | 42.45% (216MB) | 33.05% (32.3MB) | 28.28% (73.5MB) |

In spite of this difference, we show both numerically (in the next section) and empirically (in Sec. 3 and 4.3) that this technique works robustly for DNNs.

To the best of our knowledge, this work is the first to demonstrate the effectiveness of chunk-based accumulation and floating point stochastic rounding towards 8-bit DNN training of large models.

**Comparison of Accumulation Techniques**    We perform numerical analysis to investigate the effectiveness of the proposed chunk-based accumulation and floating point stochastic rounding schemes. Fig. 3(b) compares the behavior of $FP16$ accumulation for different rounding modes and chunk sizes. A vector with varying length drawn from the uniform distribution (mean=1, stdev=1) is accumulated. As a baseline, accumulation in $FP32$ is shown where the accumulated values increase linearly with vector length, as the addend has a non-zero mean. A typical $FP16$ accumulation with the nearest rounding (i.e., ChunkSize=1) significally suffers swamping errors (the accumulation stops when $length \geq 4096$, since the magnitudes differ by $\geq 2^{11}$). Chunk-based accumulation dramatically helps compensate this error, as the effective length of accumulation is reduced by chunk size to avoid swamping (ChunkSize=32 is already very robust, as shown in Fig. 3(b)). The figure also shows the effectiveness of the stochastic rounding; although there exists slight deviation at large accumulation length due to the rounding error, stochastic rounding consistently follows the FP32 result.

Given these results on simple dot-products, we employ chunk-based accumulation for Forward/Backward/Gradient GEMMs, using the reduced-precision dot-product algorithm described in Fig. 3(a). For weight update AXPY computations, it is more natural to use stochastic rounding, since the weight gradient is accumulated into the weight over mini-batches across epochs, unlike dot-product of long vectors in GEMM. The following sections empirically demonstrate the effectiveness of these two techniques over a wide spectrum of DNN training models and datasets.

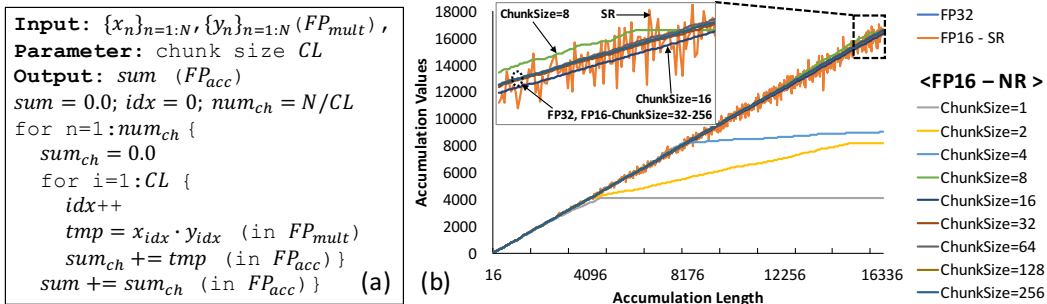

Figure 3: (a) Reduced-precision dot-product based on accumulation in chunks. (b) Comparison of accumulation for different chunk sizes and rounding modes. A typical $FP16$ accumulation (i.e., ChunkSize = 1) with nearest rounding (NR) suffers significant error, whereas ChunkSize >= 32 help compensate this error. Stochastic rounding schemes also follows the $FP32$ baseline.

# 3   Experimental Results

Reduced-precision emulated experiments were performed using NVIDIA GPUs. The software platform is an in-house distributed deep learning framework [7]. The three GEMM computations share the same bit-precision and chunk-size: $FP8$ for input operands and multiplication and $FP16$ for accumulation with a chunk-size of $64$. The three AXPY computations use the same bit-precision, $FP16$, using floating point stochastic rounding. To preserve the dynamic range of the back-propagated error with small magnitude, we adopt the loss-scaling method described in [16]. For all the models tested,

a single scaling factor of 1000 was used without loss of accuracy. The GEMM computation for the last layer of the model (typically a small FC layer followed by Softmax) is kept at $FP16$ for better numerical stability. Finally, for the ImageNet dataset, the input image is represented using $FP16$ for the ResNet18 and ResNet50 models. The technical reasons behind these choices are discussed in Sec. 4 in more detail.

To demonstrate the robustness as well as the wide coverage of the proposed $FP8$ training scheme, we tested it comprehensively on a spectrum of well-known Convolutional Neural Networks (CNNs) and Deep Neural Networks (DNNs) for both image and speech classification tasks across multiple datasets; CIFAR10-CNN ([14]), CIFAR10-ResNet, ImageNet-ResNet18, ImageNet-ResNet50 ([9]), ImageNet-AlexNet ([15]), BN50-DNN ([18]) (details on the network architectures can be found in the supplementary material). Note that, for large ImageNet networks, we skipped some pre-processing steps, such as color and scale augmentations, in order to accelerate the emulation process for reduced-precision DNN training, since it needs large computing resources.

All networks are trained using the SGD optimizer via the proposed $FP8$ training scheme without changes to network architectures, data pre-processing, or hyper-parameters, then the results are compared with the $FP32$ baseline. The experimental results are summarized in Table 1, while the detailed convergence curves are shown in Fig. 4. As can be seen, with the proposed $FP8$ training technique, every single network tested achieved almost identical test errors compared to the full-precision baseline while memory foot-print for not only weight but also the master copy is reduced by $2\times$ due to $FP8$ weight and $FP16$ master copy. As a proof of wide-applicability, we additionally trained the CIFAR10-CNN network with the ADAM optimizer [12] and achieved baseline accuracies while using $FP8$ GEMMs and $FP16$ weight updates. Overall, our experimental results indicate that training with $FP8$ representations, $FP16$ accumulations and $FP16$ weight updates show remarkable robustness across a wide spectrum of application domains, network types and optimizer choices.

Table 2 shows a comparison of the reduced-precision training work for top-1 accuracy (%) of AlexNet on ImageNet. The proposed $FP8$ training scheme achieved equivalent accuracies to the previous state-of-the-art, while using only half of the bit-precision for both representations and accumulations.

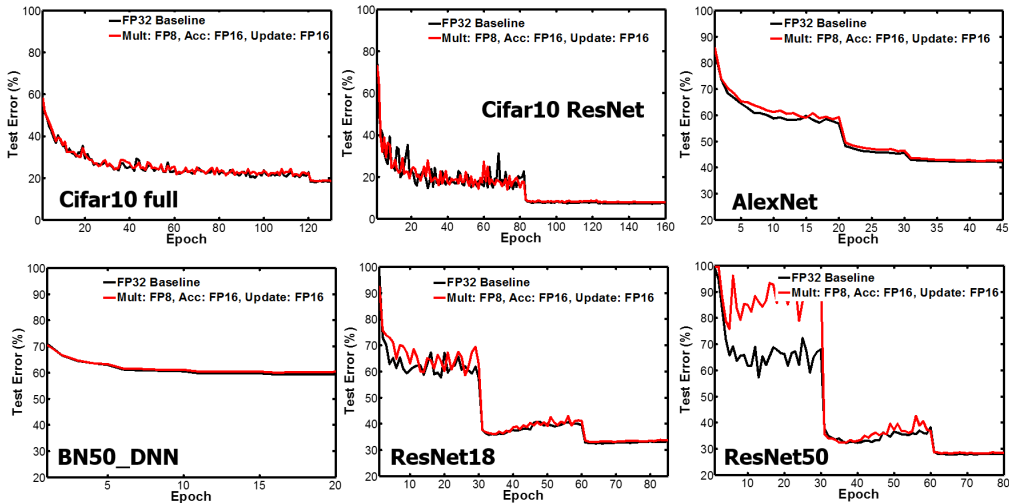

Figure 4: Reliable model convergence results across a spectrum of models and datasets using a chunk size of 64. $FP8$ is used for representations and $FP16$ is used for accumulation and updates.

## 4 Discussion & Insight

### 4.1 Bit-Precisions for First and Last Layer

The first and last layers in DNNs are often excluded from quantization due to their sensitivity [22, 3]. However, there is very limited understanding on how the bit-precision needs to be set for the first/last

Table 2: Comparison of reduced-precision training for top-1 accuracy (%) for AlexNet (ImageNet)

| Reduced Precision Training Scheme | Bit-Precision | | | | | $FP32$ | Reduced Precision |
|---|---|---|---|---|---|---|---|
| | $W$ | $x$ | $dW$ | $dx$ | $acc$ | | |
| DoReFa-Net [22] | 1 | 2 | 32 | 6 | 32 | 55.9 | 46.1 |
| WAGE [19] | 2 | 8 | 8 | 8 | 32 | N/A | 51.6 |
| DFP [4] | 16 | 16 | 16 | 16 | 32 | 57.4 | 56.9 |
| MPT [16] | 16 | 16 | 16 | 16 | 32 | 56.8 | 56.9 |
| **Proposed *FP8* training** | **8** | **8** | **8** | **8** | **16** | **58.0** | **57.5** |

Table 3: Comparison of the precision setting on the last layer of AlexNet

| Last Layer GEMMs | | | Input to Softmax | Test Error (%) | Accuracy Degradation (%) |
|---|---|---|---|---|---|
| Forward | Backward | Gradient | | | |
| FP16 | FP16 | FP16 | FP16 | 42.30 | 0.34 |
| FP8 | FP8 | FP8 | FP8 | 52.12 | **10.16** |
| FP8 | FP8 | FP8 | **FP16** | 42.37 | **0.41** |

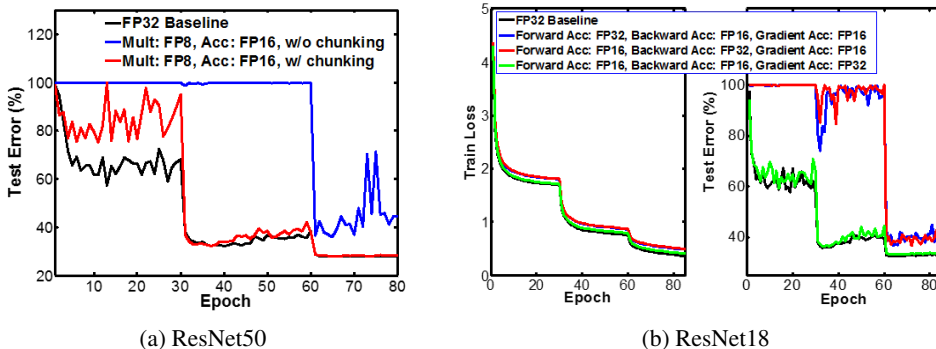

(a) ResNet50                    (b) ResNet18

Figure 5: (a) The importance of chunk-based accumulations for ResNet50. (b) Sensitivity of Forward, Backward and Gradient GEMMs to accumulation errors for ResNet18 **without chunking** - indicating that Gradient GEMM accumulation errors harm DNN convergence.

layers in order to reduce its impact on model accuracy. This section aims to precisely provide that insight and specify how the bit-precision of the first and last layers affects $FP8$ training performance.

For the first layer, we observe that the representation precision for input images is very critical for successfully training $FP8$ models. Image data is typically represented by 256 color intensity levels (e.g., uint8 in CIFAR10). Since $FP8$ does not have enough mantissa bits to represent integer values from 0 to 255, we chose to use $FP16$ to adequately represent input images. This is particularly critical for achieving high accuracy on ImageNet using ResNet18 and ResNet50; without which we observe $\sim 2\%$ accuracy degradation for these networks. All other data types includings weights, output activations, and weight gradients can still be represented in $FP8$ with no loss in accuracy.

Additionally, we note that the last layer is very sensitive to quantization. First, we conjecture that this sensitivity is directly related to the fidelity of the Softmax function (since these errors get exponentially amplified). To verify this, we conducted experiments on AlexNet, with varying precisions for the last layer. As summarized in Table 3, the last layer with all three GEMMs in $FP16$ achieves baseline accuracy (degradation $< 0.5\%$), but the $FP8$ case exhibits noticeable degradation. We also observe that it is indeed possible to use $FP8$ for all three GEMMs in the last layer and achieve high accuracy – as long as the output of the last layer Forward GEMM is preserved in $FP16$. However, to ensure robust training across a diverse set of neural networks, we decided to use $FP16$ for all three GEMMs in the last layer. Given the limited computational complexity in the last layer of a DNN ($< 1\%$ in FLOPS), we anticipate very little loss in performance from running this layer in $FP16$ while maintaining $FP8$ for the rest of layers in DNNs.

Table 4: Impact of the rounding mode used in $FP16$ weight updates. Top-1 accuracy for AlexNet and ResNet18 on the ImageNet dataset is reported for nearest as well as stochastic rounding approaches.

| | $FP32$ Baseline | Nearest Rounding | Stochastic Rounding |
|---|---|---|---|
| AlexNet | 58.04% | 54.10% | 57.94% |
| ResNet18 | 67.43% | 65.74% | 67.34% |

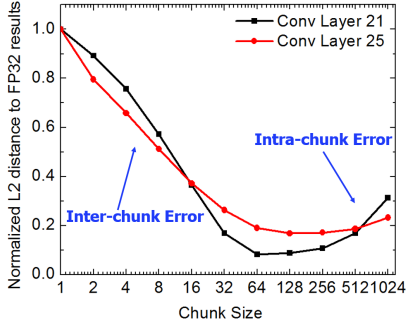

Figure 6: Effect of chunk sizes on Gradient GEMM computation errors (normalized L2-distance between $FP8$ and $FP32$ GEMMs) for CIFAR10-ResNet.

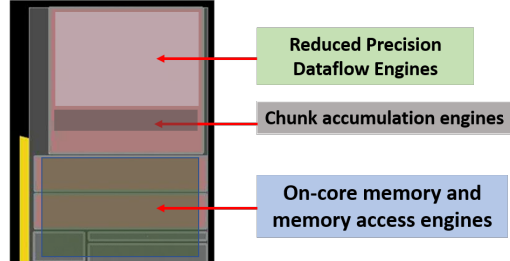

Figure 7: Chip layout of a novel dataflow-based core (14 nm) with $FP16$ chunk-based accumulation. $FP8$ engines are $> 2 \sim 4\times$ more efficient over $FP16$ implementations - and require lesser memory bandwidth and storage

## 4.2 Accumulation Error

Next, we investigate the impact of chunk-based accumulations. Prior works (e.g., [16, 4]) claim that 32 bits of precision is required for the accumulation in any GEMM to prevent loss of information. Motivated by the significant area/energy expense of $FP32$ adders, we counter this by claiming that chunk-based accumulations can effectively address this loss in long dot-products while maintaining accumulation bit-precision in $FP16$. As shown in Fig. 5(a), $FP8$ training for ResNet50 fails to converge without chunking, but chunk-based computations bring model convergence back to baseline.

Investigating further, we identify Gradient GEMM to be the most sensitive to accumulation precision when chunking is not used. As shown in Fig. 5(b), $FP8$ training on ResNet18 converges to baseline accuracy levels when $FP32$ is used for Gradient GEMM. For other cases, interestingly, the training loss converges but the test error diverges to 99%, exhibiting significant over-fitting. This implies that the failure in addressing information loss in low-precision Gradient GEMM results in poor generalization of the network during training. Gradient GEMM accumulates weight gradients across minibatch samples, where information from small gradients may be lost due to swamping (Sec. 2.3), resulting in the SGD optimization being stuck at sharp local minimas. Chunk-based accumulation addresses the issue of swamping to recover information loss and therefore help generalization.

To understand the impact of chunk size on accumulation accuracy for Gradient GEMM in DNN training, we extracted data from Activation and Error matrices from the two different Conv layers in the CIFAR10-ResNet model to compute Gradient GEMM with varying chunk sizes. Fig. 6 shows the normalized L2-distance of the results relative to the full-precision counterpart for varying chunk size. The computation results are closest to the $FP32$ baseline with the chunk size between 64 and 256. Before and after this range, the L2-distance is higher due to the dominant inter-chuck and intra-chunk accumulation error, respectively. Based on this insight, and for the ease of hardware implementation, we use a chunk size of 64 for our experiments across all models.

## 4.3 Nearest Rounding vs. Stochastic Rounding

Finally, we investigate the impact of rounding mode on $FP16$ weight updates. Since weight gradients are typically several orders of magnitude smaller than weights, prior work (e.g., [16]) adopts $FP32$ for weight updates. In this work, we maintain $FP16$ for the entire weight update process in SGD (i.e., L2-Reg, Momentum-Acc, and Weight-Upd), as a part of our $FP8$ training scheme; stochastic rounding is applied to avoid accuracy loss. Table 4 shows the impact of rounding modes (nearest vs.

stochastic) on the top-1 accuracy of the AlexNet and ResNet18 models. For this experiment, GEMM is done in $FP32$ to avoid its additional impact on accuracy. As can be seen from the table, the nearest rounding suffers noticeable accuracy degradation ($2 \sim 4\%$) while stochastic rounding maintains the baseline accuracies, demonstrating its effectiveness as a key enabler for low precision training.

### 4.4 Hardware Benefits

A subset of the new ideas discussed in this paper were implemented in hardware using a novel dataflow based core design in 14nm silicon technology – incorporating both chunk-based computations as well as scaled precisions for training (Fig. 7). Through these hardware implementations, we draw the following conclusions: 1) The energy overheads of chunk-based computations are < 5% for chunk sizes > 64. 2) $FP8$ based multipliers accumulating results into $FP16$ are 2-4 times more efficient in hardware compared to pure $FP16$ computations because of smaller multipliers (i.e., smaller mantissa) as well as smaller accumulator bit-widths. $FP8$ hardware engines are roughly similar in area and power to 8-bit integer computation engines (that require larger multipliers and 32-bit accumulators). These promising results lay the foundation for new hardware platforms that provide significantly improved DNN training performance without accuracy loss.

## 5 Conclusions

We have demonstrated DNN training with 8-bit floating point numbers ($FP8$) that achieves $2 - 4\times$ speedup without compromise in accuracy. The key insight is that reduced-precision additions (used in partial product accumulations and weight updates) can result in swamping errors causing accuracy degradation during training. To minimize this error, we propose two new techniques, chunk-based accumulation and floating point stochastic rounding, that enable a reduction of bit-precision for additions down to 16 bits – as well as implement them in hardware. Across a wide spectrum of popular DNN benchmarks and datasets, this mixed precision $FP8$ training technique achieves the same accuracy levels as the $FP32$ baseline. Future work aims to further optimize data formats and computations in order to increase margins as well as study additional benchmarks and datasets.

**Acknowledgments**

The authors would like to thank I-Hsin Chung, Ming-Hung Chen, Ankur Agrawal, Silvia Melitta Mueller, Vijayalakshmi Srinivasan, Dongsoo Lee and Jinseok Kim for helpful discussions and supports. This research was supported by IBM Research, IBM SoftLayer, and IBM Congnitive Computing Cluster (CCC).

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
