[Supplementary Material]

# Supplementary Materials for Training Deep Neural Networks with 8-bit Floating Point Numbers

**Naigang Wang, Jungwook Choi, Daniel Brand, Chia-Yu Chen and Kailash Gopalakrishnan**
IBM T. J. Watson Research Center
Yorktown Heights, NY 10598, USA
`{nwang, choij, danbrand, cchen, kailash}@us.ibm.com`

## 1   Network Architectures

- CIFAR10-CNN [2]: 3 Conv layers (with 5x5 filters and ReLU activation function), 1 FC layer, and a 10-way Softmax.
- CIFAR10-ResNet [1]: 15 ResNet blocks totaling 31 convolutional layers with 3x3 filters, batch normalization, ReLU activation and a final FC layer with a 1K Softmax.
- AlexNet [3]: 5 Conv layers and 3 FC layers. The output layer is a 1K Softmax.
- ResNet18 [1]: 8 ResNet blocks totaling 16 Conv layers with 3x3 filters, batch normalization, ReLU activation and a final FC layer with a 1K Softmax.
- ResNet50 [1]: 16 bottleneck ResNet blocks totaling 48 Conv layers with 3x3 or 1x1 filters, batch normalization, ReLU activation and a final FC layer with a 1K Softmax.
- BN50-DNN [5]: 6 FC layers (440x1024, 1024x1024, 1024x1024, 1024x1024, 1024x1024, 1024x5999) and a 5999-way Softmax.

## 2   Datasets

- The CIFAR10 dataset [2] is an image classification benchmark containing 32x32 pixel RGB images. It consists of 50K training and 10K test image sets.
- The ImageNet dataset [4] is an image classification benchmark which consists of 1000-categories of objects with over 1.2M training and 50K validation images. Images are first resized to 256x256 and then randomly cropped to 224x224 prior to being used as input to the network.
- The BN50 dataset [5] is a speech recognition benchmark which is based on the English Broadcast News (BN) training corpus and containing a 45-hour training set and a 5-hour hold out set.