[Reviews · NeurIPS 2018]

Reviewer 1



The main goal of this work is to lower the precision of training with deep neural networks to better use new styles of hardware. In particular, the technical idea is to lower the size of the accumulator in dot products and the representation of all weights. The main observation is that a form of chunking, which is well known in the optimization and math programming community, has sufficiently better error properties to train DNNs. The paper is clear, simple, and effective. - The techniques are standard in this area, and so it's unclear if its met the novelty bar for NIPS. - A more thorough citation of the work on reduced precision could help better position the contribution. - Claimed contributions appear to be overstated or under validated. The title is a bit misleading, since it really is a mixed precision paper as a result of many of the numbers actually being FP16--not 8. And it's not about all of deep learning--just CNNs and while this is substantial enough to be interesting, it's not a general purpose deep learning solution. FP8 format contribution. There is no experimental support for the particular choice of FP8 (one could choose other mantissa/exponent combinations). As this is claimed as a contribution but not justified in experiments. The claim of designing the format is also a bit of a stretch here: unless, I've missed something, it's just filling out the parameters to the standard FP model (which gives one all of the analytic results on blocking from below). I think the paper could be helped by more careful positioning of the contributions, both techniques are well known in this area: Reduced precision with stochastic rounding has been used by several authors in this area as has blocking. More below. While phrased in terms of dot products, the core issue here is the numerical stability of sums. This is very well studied in the HPC literature. There is a single citation narrowly around dot products [1], but this has been well studied and is often used in intro courses. The novelty lies in its application to deep learning and showing that this simple technique can obtain good error properties. This work does not subtract from the contribution of this paper, but I think would help give context about the differing role of stochastic rounding. A representative example is QSGD from Alistrah et al in NIPS17. Other communities including the architecture community have picked up on these lines of work (see recent ISCA or MICROs). A more full accounting of related work would help. That said, these works have focused on stochastic style rounding as the only technique--neglecting the nice insight here about the effectiveness of blocking. This is well known technique in the HPC community, and is an even simpler version of the blocking technique. The validation is light to make the claim about all deep learning as it only examines CNN. If the paper have claims about other neural networks that would strengthen the paper. No details of the hardware were provided. This is probably unavoidable. Author feedback: I believe folks are excited, and I think adding in more related work would strengthen the paper. It's exciting work.

Reviewer 2



The work at hand demonstrates that it is, given certain modifications, possible to train DNNs using 8-bit floating point numbers (instead of the commonly used 16- and 32-bit representation). Section 1 motivates the training of DNNs using only 8-bit representations and also sketches the decrease w.r.t. the test error in case of reduced precisions (1-2%, see Figure 1). Section 2 describes the modifications made by the authors. In particular, the authors show that one can successfully train the networks using 8-bit representations for all of the arrays used in matrix/tensor operations when using 16-bit sums for accumulating two 8-bit operands. This is achieved using two novel schemes, chunk-based accumulation and floating point stochastic rounding, which help to prevent swamping. Section 3 provides an experimental evaluation, which indicates that one can successfully train a variety of networks using a reduced precision without a decrease in model accuracy. Section 4 provides a detailed discussion of the results achieved. Conclusions are drawn in Section 5. The paper is well written and addresses an interesting and challenging problem. While this is not precisely an area of my expertise, I have the feeling that the work at hand depicts a solid piece of work, which might influence future implementations of DNNs: The problem is motivated nicely and the modifications made seem to make sense and are justified experimentally. The final results (Figure 4) also look promising and indicate that one basically further reduce the representation to 8/16 bits in the context of training DNNs. This will potentially lead to a speed-up of 2-4 in future. The authors have also implemented their modifications in hardware. To sum up, I think this depicts a solid piece of work, which might lay the foundation for new hardware platforms (as stated by the authors). Due to the importance of such implementations, I think this work is of interest for the NIPS community. Minor comments: line 120: chunk-based line 161: url to sharelatex project Table 2: FP8 not in bold letters

Reviewer 3



This manuscript provides a clear step forward in low precision training, through a mixture of novel ideas and solid experimental results. To summarize the paper: 1. Propose a new data format, or rather, recipe for combining data formats, that allows for training with a majority of the operations performed in an 8-bit floating point format. This includes some novel tricks like chunk-based accumulation 2. Show comprehensive experiments on Cifar10-ResNet and ImageNet1k-AlexNet to validate the method. 3. Demonstrate in silicon that that these methods are useful for optimized deep learning hardware. Overall it's a solid paper, and there are lots of things I like about it. The authors clearly understand the challenges of low precision training, and are familiar with tricks of the trade such as stochastic rounding and loss scaling. Figure 1 provides a great summary of the 3 major issues that practitioners encounter in low precision training, disentangling accumulation, weight update and storage of weights, activations and gradients. It's commendable that the authors take the space to explain the basics like this even in the 8 page NIPS format. The same can be said about Figure 2, it's a great way to succinctly capture all of the operations and what precision they are in. There are no major flaws in the manuscript, but a few minor points that could use clarification: Why was the (1,5,2) format chosen for fp8 and (1,6,9) for fp16? The 5 exponent bits for fp8 might seem natural, since it's the same as IEEE fp16, but did the authors explore e.g. (1,4,3) or (1,6,1) and find them to work less well? If IEEE (1,5,10) does not have enough dynamic range, wouldn't (1,7,8) do even better? Figure text is too small and unreadable in print or of PDF viewers that don't easily allow zooming in. I realize space is at a premium, but I would consider removing Figure 7 and section 4.4, as it's independent of the rest of the paper, and I am sure there is a separate publication in the works that will going into a great deal more detail on the hardware. Otherwise consider shortening the introduction, or removing some of the detail on Stochastic Rounding. Line 140 onwards provides a useful refresher, but the results in Table 4 and section 4.3 are not novel and could be removed / shortened. Minor comments - I was not familiar with the term "Swamping", is there a reference for where this expression comes from? - 161 seems to have a stray link - 192, does ResNet18, 50 refer to the i1k sized network, or CIFAR? Please clarify. - Table 2, what does the column "FP32" mean? Is this the accuracy reported by various authors for training the same fp32 model? Should they not all be the same? - Figure 4, ResNet50 test error is around 29%, but should be closer to 24% based on He 2016. Benchmarks such as DawnBench and MLPerf generally require 25% or better on ResNet50. Can you describe differences between this ResNet and the original one and explain the discrepancy? - 253, typo, dominated -> dominant ——- Update after rebuttal: I don’t have much to add in response to the rebuttal, but would like to urge the authors to incorporate the suggestions from Reviewer 1, which I found very valuable.